# Profiling Analysis of Filter Feeder *Polypedilum* (Chironomidae) Gut Contents Using eDNA Metabarcoding Following Contrasting Habitat Types—Weir and Stream

**DOI:** 10.3390/ijerph191710945

**Published:** 2022-09-02

**Authors:** Boobal Rangaswamy, Chang Woo Ji, Won-Seok Kim, Jae-Won Park, Yong Jun Kim, Ihn-Sil Kwak

**Affiliations:** 1Fisheries Science Institute, Chonnam National University, Yeosu 59626, Korea; 2Department of Ocean Integrated Science, Chonnam National University, Yeosu 59626, Korea

**Keywords:** chironomid gut contents, eDNA metabarcoding, next-generation sequencing (NGS), river and weir habitats

## Abstract

We analyzed the dietary composition of *Polypedilum* larvae among two contrasting habitats (river and weir). Our approach was (i) to apply eDNA-based sampling to reveal the gut content of the chironomid larvae, (ii) the diversity of gut contents in the two aquatic habitats, and (iii) assessment of habitat sediment condition with the food sources in the gut. The most abundant food was Chlorophyta in the gut of the river (20%) and weir (39%) chironomids. The average ratio of fungi, protozoa, and zooplankton in river chironomids gut was 5.9%, 7.2%, and 3.8%, while it was found decreased to 1.2%, 2.5%, and 0.1% in weir chironomids. Aerobic fungi in river midge guts were 3.6% and 10.34% in SC and IS, while they were in the range of 0.34–2.58% in weir midges. The hierarchical clustering analysis showed a relationship of environmental factors with food contents. Abiotic factors (e.g., pH) in the river and weir habitats correlated the clustered pattern with phytoplankton and minor groups of fungi. This study could help understand the food source diversity in the chironomid and habitat environmental conditions by using eDNA metabarcoding as an effective tool to determine dietary composition.

## 1. Introduction

Non-biting midges (Diptera: Chironomidae) are one of the most abundant invertebrates, widely distributed in lentic and lotic environments. The larvae of chironomid are important constituents in freshwater systems [1]. As Vasquez et al. (2022) [2] mentioned, chironomids are the biological indicators of aquatic health, the prevalence of ecological organization, functions, and systematics. Chironomids are sensitive to extreme conditions and their presence represents quality factors of the environment [3]. Genus *Polypedilum* is well known for its physiology among the Chironomidae family. Being omnivores, chironomids have a considerable mode of flexibility in their feeding and there are certain factors, such as resource availability, food quality, and physiochemical parameters, that influence their feeding behavior [4]. Analysis of larval gut content provides a descriptive illustration of the presence, existence, and interactions of indigenous natural communities. However, gut content using optical observations in chironomid is relatively time consuming and requires enormous morphological identification expertise to be accurate when compared to eDNA sequencing methods.

Earlier research reported microscopic examination-based gut content analysis without information on the multi-compositional diet [5]. Lemes-Silva et al. (2014) [6] could distinguish the food contents as organic matter, filamentous algae, and microalgae only to the lowest taxonomic level in chironomids. A similar analysis, reported by Butakka et al. (2016) [7], could categorize the food content as algae, fungal spores, and plant fragments but there was inadequacy in diverse composition and high-taxonomic resolution. Certain observations could identify phytoplankton [5,8], mostly algae [9,10] and detritus [11,12], as the major gut content in the chironomid. However, these results could not profile detailed taxonomic specificity and diversity of food sources. eDNA metabarcoding altered the perspective in terms of sampling range and increased resolution of taxonomic identification targeting different species with wider community functionalities. Further, eDNA metabarcoding has been successfully applied to demonstrate the status of rare and threatened animals, including amphibians, mammals, insects, fish, and crustaceans, in the freshwater ecosystem [13]. Recently, the technique has been employed to identify the biomass of fish species in marine habitats [14], diversity in river systems [15], and also in enumerate phyla, representing different birds, plants, and invertebrates from the terrestrial ecosystem [16].

The 18S rRNA gene has been used in eDNA metabarcoding mostly for its extensive coverage of the eukaryotic domain. The 18S V9 region [17], relatively 240 bp short fragments are the scalable feature to reveal most of the micro-eukaryotic diversity. Cordier et al. (2022) [18] explored the assemblage of 0.24 million eukaryotic ASVs in 1.95 billion DNA reads, representing the plankton diversity in deep-ocean sediments throughout the world. The 18S V9 primers showed potency to locate different species exclusively from the larger assemblage group to the minor fractions, including novel species of phytoplankton, fungi, and zooplankton [19,20,21] and different faunal communities [22,23,24]. Gut content profiles using eDNA analysis have dramatically increased in the context of taxonomic diversity ecology. Dietary constituents from eDNA are a crucial research strategy to determine the resources in the habitat and biodiversity in the ecosystem [25]. Meta-analysis of gut composition has been discussed in crustaceans, copepods, and other aquatic animals [25,26,27,28], while only a few works have been reported on the chironomid diet. The initial works of Jo et al. (2020) [29] on eDNA sampled 18S rRNA metagenomic sequencing indicated the food content of chironomids with different phytoplankton at the species level. This study represents the first eDNA-based diet composition information among generalist *Polypedilum* midges. We aimed to analyze and identify the different gut contents of the *Polypedilum* sp. between river stretches and large-scale weirs. eDNA metabarcoding was used to explore the dietary profile of midges based on the hypothesis: (i) to identify the diet profile of the larvae in both habitats, (ii) to scrutinize different food contents by comparing both the study habitats, and (iii) to assess the relationship between food content and the habitat condition.

## 2. Materials and Methods

### 2.1. Field Sampling and Measuring Factors

The study areas selected were the major water bodies located in the region of South Korea (Figure 1). The portion of samples collected from the river Sunchang (SC) and Imsil (IS), the main streams of the region, and the samples from four largescale weirs denoted Ipo (IP), Sejong (SJ), Juksan (JS), and Gangjeong (GG). The gut samples were given the sample code of the sites where the midges were collected. The sampling sites exhibit a complex hydrological dynamic relating to different environments with geographical features.

The sampling was carried out quarterly from April 2019 to July 2019. Triplicates of samples were collected vertically at depths of 10–50 m in weirs and less than 1 m in the river. Water quality parameters, pH, water temperature (°C), conductivity (µS/cm), turbidity (NTU), and dissolved oxygen (DO, mg/L) were measured on site using portable equipment (Professional Plus, YSI, Yellow Springs, OH, USA). In the laboratory, water samples were filtered through a 0.45 µm pore-size membrane (Advantec MFS membrane filter, Irvine, CA, USA) for water nutrients. An automatic water quality analyzer (AutoAnalyzer 3 HR, Seal Analytical Inc., Mequon, WI, USA) was used for the estimation of total phosphorus, total nitrogen, and Chlorophyll-a concentration, and the measurements of optical density were performed using a UV spectrophotometer (GENESYS™, Thermo Fisher Scientific, Waltham, MA, USA). Dissolved organic carbon (DOC) and total organic carbon (TOC) concentrations were measured using a TOC analyzer (Vario TOC cub, Langenselbold, Germany) and determined through an 850 °C combustion catalytic-oxidation method [24]. Further, 4th instar larvae of Chironomidae, *Polypedilum* sp. were collected using Surber net (25 cm × 20 cm), dredging (1 m × 1 m), Ekman grabs, and Ponar grab. The larvae samples obtained were preserved immediately in 96% ethanol and maintained at 4 °C for DNA meta-barcoding analysis.

### 2.2. DNA Extraction and Metagenomic Sequencing

The gut samples were dissected from the 4th instar *Polypedilum* sp. larvae of each sample site followed by a sample processing procedure after the complete volatilization of ethanol (Jo et al., 2020 [29]). Genomic DNA was extracted using DNeasy Blood & Tissue Kit (Qiagen, Düsseldorf, Germany) as per the manufacturer’s protocol. The quality and integrity of the gDNA extracted were measured using PicoGreen (Thermo Fisher Scientific, Waltham, MA, USA), VICTOR Nivo Multimode Microplate Reader (PerkinElmer, Waltham, MA, USA) and were prepared for sequencing according to Illumina 18S Metagenomic Sequencing Library protocols (San Diego, CA, USA).

The initial thermal cycle corresponds to the amplification of the adapter region using the condition 95 °C for 3 min followed by 25 cycles of 95 °C for 30 s, 55 °C for 30 s, 72 °C for 30 s, and a final extension of 5 min at 72 °C was carried out using the 18S V9 primers including an adapter sequence (Table 1). The second set of amplification as an indexing PCR was performed using the previous amplicon using conditions as follows: 95 °C for 3 min; 8 cycles at 95 °C for 30 s, 55 °C for 30 s, 72 °C for 30 s, and extension step for 5 min at 72 °C. The final products were normalized to the concentration of DNA and then pooled using PicoGreen (Thermo Fisher Scientific Waltham, MA, USA) and also the size of the libraries was verified using LabChip GX HT DNA High Sensitivity Kit (PerkinElmer, Waltham, MA, USA). The amplicon libraries were sequenced using the MiSeq™ NGS platform (Illumina, San Diego, CA, USA).

### 2.3. Bioinformatics Analysis of the Sequence Reads

Contiguous sequences were created from the sequence read data using the ‘make.contigs’ command of Mothur Miseq SOP (mothur, v.1.47.0; https://www.mothur.org/wiki [30]). Low-quality sequences that contained any ambiguities, homopolymer runs of a length ≥ 8 bp, and sequences < 275 bp for 18S V9 rRNA were removed using the ‘screen.seqs’ command. The reference database for the specific hypervariable region for taxonomic identification was prepared from the SILVA reference database, v.138.1 [31] using the ‘pcr.seqs’ command and the alignment was attained using ‘align.seqs’ routine. Pre-clustered sequences were checked for chimeric regions using VSEARCH [32] and we removed such sequences from the analysis. The custom reference database mentioned above was used for the taxonomic assignment of the sequences. Taxonomic classification was performed using the ‘classify.seqs’ command and the unknown alignments were removed following further quality control steps. Pairwise distances were calculated (‘dist.seqs’) and OTUs clustered (‘cluster’) using the distance threshold of 0.03. OTU-based alpha and beta diversity measurements proceeded after the classifying taxonomy towards OTUs, summarizing for analysis, respectively. A two-way and four-way Venn diagram was created to visualize the OTUs shared between each combination of the regions (https://bioinfogp.cnb.csic.es/tools/venny/index.html; accessed on 13 June 2022 [33]).

### 2.4. Statistical Analysis of OTUs and Environmental Factors

OTUs were defined by Mothur (v.1.47.0; Schloss et al., 2009 [30]) using a 0.03 cutoff distance. The number of reads per sample was randomly subsampled to reduce sequence bias, respectively. Using Mothur (v.1.47.0), Good’s coverage, alpha diversities including Shannon index [34], and Chao richness [35] were calculated.

The relative frequency of the taxa with assigned OTUs in river and weir habitats was calculated and used to describe the diet composition of the chironomid. We calculated the diet breadth
(1)$$B={\left(\overset{n}{\underset{i=1}{\sum }}p_{i}^{2}\right)}^{-1}$$
using Levin’s index (1968) [36] where $p_{i}$ is the proportion of food items i and n is the total number of food items in the gut content. Standardized niche measures ranging from 0 to 1 were applied using Hurlbert’s formula (1978) [37]:(2)$$B_{A}={\left(n-1\right)}^{-1}\;\left({\left(\sum_{i=1}^{n}p_{i}^{2}\right)}^{-1}-1\right)$$
where $p_{i}$ is the proportion of food sources i  in the diet and n is the total food content.

To demonstrate the relationship between the environmental condition and food contents in the habitats we used the hierarchical clustering method [38]. The variables were grouped using Ward (1963) [39] method after the ordination of beta diversity between the relative OTUs of food items at the phylum level and environmental variables using Bray–Curtis dissimilarity [40]. The two-way clustering analysis was performed using the vegan package in R Statistical Software (v4.1.3; R Core Team 2022 [41]).

## 3. Results

### 3.1. Diversity of Polypedilum Larvae Gut Content Using V9 Metabarcoding

The gut content of *Polypedilum* larvae (Chironomidae) in the river and weir habitats was demonstrated using OTU analysis. OTUs were assigned by calculating the 0.03 cut-off distance. Total eukaryotic OTUs detected were 4066 in SC, 6361 in IS among river chironomid and it was found to be 2004 OTUs in IP, 2031 in SJ, 2014 in JS, and GG with 1372 OTUs, respectively. Taxonomic classification using the SILVA reference database revealed the composition of the food content. As such, 24 phyla (class (54), order (89), family (105), genus (140)) from river chironomid and 25 phyla (class (62), order (104), family (125), genus (188)) from weir chironomid gut contents were identified. Altogether, we grouped 29 phyla into phytoplankton, protozoa, fungi, zooplankton, and a few minor OTU taxa as others.

The highest number of OTUs detected was associated with Chlorophyta, Diatomea, and Ochrophyta (Figure 2). River chironomid gut profile was constituted of 60% phytoplankton, representing eight different taxa in IS (36%) and five taxa in SC (24%). Further, 21–23% of the weir midge diet composition, except for GG (15%), represented phytoplankton (82.4%). Among the OTUs assigned to phytoplankton-microalgae (diatomea), Ochrophyta-golden algae (Chrysophyceae) and green algae (Chlorophyta) were the dominant groups in the river chironomid gut profile while the green algae (Chlorophyta), microalgae (diatomea), and Ochrophyta occupied the prominent clusters of weir midge gut profiles. The phylum diatomea was assigned to 13% OTUs in IS, which was double the proportion of SC and it was 12% (IP), 9% (SJ), and 6% (JS and GG) in the weir chironomid. The total composition of the group Ochrophyta was 16% in river chironomid (8% in SC and IS) and with a range of 5% in IP among weir samples. The majority of OTUs for microzooplankton identified in the gut samples was assigned to Cercozoa and Euglenozoa in the river habitants’ diet while it was the least represented OTUs of diet in weir chironomid, including other taxa, such as Ciliopora, Heterolobosea, Protalveolata, and Amoebozoa. Approximately 12% of the food contents in river chironomids was delineated to protozoa, whereas it was 4% in the weir gut samples. Mesozooplankton in the diet profile was associated with Rotifera, Annelida, Arthropoda, Mollusca, Cnidaria, and Nematozoa. The maximum OTUs among zooplankton were assigned to Rotifera with 7% in river midges (SC > IS) and <1% in weir midges. In the gut profiles of both the river samples and specifically in JS, GG (weir samples) Nucleariidae/Fonticula group, an unclassified eukaryote was identified. Likewise, Euryarchaeota and Halobacterota were identified as rare groups clustered in JS. Moreover, the taxa of minor OTUs with <1% coverage were configured as others.

According to the OTU-based analysis, as in Table 2, the gut profile of IS was found to be more diverse than SC among river chironomid and also more diverse than the weir profile. Levin’s standardized niche breadth corresponds to diet breadth (B), determined from the OTUs of the river and weir chironomid diet. Niche breadth (B_A_) was found to be low and relatively proximal values, though the maximum was recorded in IS (0.27) in the river gut profile. Among the weir midge diet profile, IP was recorded with 0.13 as the maximum value. IS chironomid gut profile was recorded with the maximum measure of Chao species richness (8023) for the abundance of each species. In the weir profile, GG represented a high species richness of about 3024. Further, 100% sample coverage was estimated in the study.

### 3.2. Composition of Dominant and Unique OTUs in the Gut Profile

Among the taxa identified by the eDNA approach, 71% of the phyla was shared in both IS and SC gut profiles (Figure 3A). Likewise, 46% of the taxa existed as common diet content in the weir gut profile (Figure 3B). Further, 17 common taxa and 25% of the unique OTUs of 6 exclusive taxa represented the diet composition of IS river chironomids whilst 4% in SC was Crytomycota as unique taxa and 46% of shared OTUs were recorded in the weir midges gut profile.

OTUs of Arthropoda are shared between IP, SJ, and JS. Rotifera was the shared food item in SJ, JS, and GG. Protalveolata was the common taxon in SJ and GG, Halobacterota in IP and JS, Ciliophora, Cnidaria, and Nucleariidae/Fonticula were the common diet components in JS and GG. Heterolobosea (IP), Annelida (SJ), Euryarchaeota, and Amoebozoa (JS) were the unique food items in the weir gut profile. The common OTUs could be considered to be the main food sources in the river and weir ecosystem. The unique OTUs could represent the diversity in the habitats.

Accounting for the probability of common OTUs, we further extended the taxonomic fingerprint analysis to exhibit dominant families. Chlorophyta groups multifurcated to Chlorophyceae, Mamiellophyceae, Trebouxiophyceae, and Ulvophyceae. Maximum food content belonged to the family Chlorophyceae in the gut samples from the rivers (13%) and weirs (22.8%). Other families were identified with ≤1% in the dietary profile, respectively. Bacillariophyceae, Coscinodiscophytina, and Mediophyceae were the subsets of the Diatomea identified in the gut content. Bacillariophyceae and Coscinodiscophytina occupied approximately 4% of the total gut profile of river chironomid, whilst it was Mediophyceae and Bacillariophyceae at around 7% in the gut content of weir habitants. The maximum ratio of phylum Ochrophyta was composed of Chrysophyceae with 4% in river midges, whereas it was Eustigmatophyceae with 2% in weir organisms. Other subsets of Ochrophyta (Xanthophyceae and Raphidophyceae) were observed with a <1% contribution to the profile. Cercozoa and Euglenozoa were the diverse groups of Protozoa in both habitat gut samples. Almost >5% of food content was defined by protozoa in the river gut profile and the weir gut profile was observed with <2% of protozoa.

### 3.3. Relationship of Environmental Factors and the Community Composition of Gut Contents

The environmental variables related to the distribution of food sources in the river and weir habitats were demonstrated using hierarchical cluster analysis (HCA, Figure 4).

The food contents were grouped as phytoplankton, protozoa, fungi (aerobic, anaerobic, and facultative anaerobic), zooplankton, and others (minor/rare OTUs). Two cluster analyses showed the correlation pattern of environmental conditions with the specific food source equivalently clustered in the habitats, shown in the dendrogram. The dominant group of phytoplankton showed an association with conductivity. To this, an equivalent correlated cluster of facultative anaerobic fungal groups was found associated with pH, DO, turbidity, total nitrogen, chlorophyll-a, and water temperature. A second cluster pattern indicated a correlation of protozoa, aerobic and anaerobic fungi, zooplanktons, and all minor OTUs (others) with total phosphorus.

### 3.4. Assessment of the Habitat Sediment Condition through Fungi Communities

Additionally, we analyzed the detailed frequency of fungi grouped as aerobic and anaerobic fungi to evaluate the habitat and sediment conditions. Feature fungal groups (aerobic and facultative anaerobic) of the profile were represented in Table 3, with 10% (IS) and 3.6% (SC) unique OTUs in the gut contents of the river chironomids. The ratio of aerobic and facultative anaerobic fungi was about 0–2% in weir chironomids. Furthermore, the anaerobic taxa, phylum Neocallimastigomycota, was observed as rare OTUs in GG. The aerobic fungal groups showed maximum lineage to phylum Ascomycota in the gut contents of larvae from river locales. The proportions of Basidiomycota were significantly less and Peronosporomycetes were not identified in SC.

## 4. Discussion

Increasing urbanization leads to habitual alteration and quality deterioration in terms of geographical features, mainly in the freshwater ecosystem [42]. Ecological studies based on eDNA sequencing technologies acquire importance due to time efficiency and high detection sensitivity [43,44]. The eDNA sampling approach increased the frequency of interest in environment analogies by decrypting the data of complex nature [45,46]. Differential composition of food items was recorded in the diet profiles. The diverse array of taxa in the gut content was obtained as an exceptional approach using eDNA metabarcoding of the 18S V9 region. This study attempted to present the dietary profile of *Polypedilum* in the river and weir habitats. The eDNA metabarcoding approach demonstrated the representatives of phytoplankton, protozoa, fungi, and zooplankton groups of varied ratios in the chironomid gut samples. According to the functional OTUs identified, the relative frequency of the diet profile was determined. The diversity of the food resource in the habitats and the diet of midges relied mostly on the phytoplankton corresponding to algae, relating the observations of Jo et al. (2020) [29]. According to the predominant groups and relative genus coverage to the assigned OTUs, we could categorize the algal groups into Chlorophyta (green algae), Diatomea (microalgae), and Ochrophyta (golden algae—Chrysophyceae). In addition, the minor OTU group (taxa) in the profile corresponded to the multi-composition diet of midges. Niche breadth related to the environmental factors persisted in the study sites, depicting the diet breadth of each food item in the gut.

### 4.1. Overview of the Gut Content and Diet Composition

As a result of *Polypedilum* gut content, algae were found to be the abundant component in gut content. Chlorophyta and Diatomea were the other two major clusters recorded in phytoplankton (Figure 2). In both river and weir chironomids, Chlorophyta was identified to be the dominant taxon, representing the different groups of green algae. Filter feeder chironomid depends upon the food available in the habitats [47]. Chlorophyta is a dominant food item [8,12]; specifically, green algae [11] and Diatomea [7] in the gut content of chironomid have also been reported in previous studies based on microscopic observation. Certainly, these groups have been reported in water sample (eDNA) based content exploring the biodiversity of the freshwater habitat [19]. Alongside the microalgal population, the phylum Ochrophyta was identified as the third most common food item in both river and weir chironomid guts. We observed outputs related to Rodríguez-Barreras et al. (2020) [48] regarding the species variation and sample composition of phylum Ochrophyta identified between study habitants. Phragmoplastophyta, Cryptophyta, Dinoflagellata, and Haptophyta were the minor OTU-associated groups recorded as phytoplankton diet sources. Gut content associated with coexisting food source community is still accorded as a future hypothesis [45]. We could configure aerobic and facultative anaerobic fungal groups in both river and weir chironomids. Further, there was a single cluster of anaerobic fungi (Neocallimastigomycota) in weir chironomids. Fungi in the chironomid diet have been reported based on optical observations [7,11] but could not provide a better resolution of taxa. However, investigations based on eDNA have reported such diverse fungal groups in the gut content of *Megaplatypus mutatus* and mammals [49,50] and different environments [51,52]. Rotifera occupied a unique proportion in the diet profile of river chironomids. Likewise, the ratio of Protozoa content in the diet was the determining factor of diet breadth associated with the OTUs of river chironomids. Similar information on diet composition with rotifers and Protozoa has been provided by other authors, indicating planktonic rotifers, Bacterivorous-Protozoa-associated chironomid diet [11], and foraminifer protozoans in a diet of small isopods [53]. Certain observations have also proposed that the higher proportion of meiofauna organisms in a system could be actively preyed upon by chironomid larvae [11,54,55]. We categorized the meiofauna community as others since this content was observed with a <1% ratio in the gut. Minor taxa were found in the diet composition; the compositional food content <1% ratio was assumed to be the ingested items like detritus portions, suspended particles in the water and bottom layers [54], considered to be of marginal importance [11].

### 4.2. Diversity of Gut Content Associated with Food Sources in Habitats

The presence of phytoplankton in the diet profile of *Polypedilum* constituted Diatomea, Chlorophyta, Ochrophyta, and Phragmoplastophyta as the major division and the Marine Stramenopile, Prymnesiophyceae, Cryptophyceae, and Dinoflagellata as the minor contents. Green algae identified in the gut content of chironomids belong to Chlorophyceae, Mamiellophyceae, Trebouxiophyceae, and Ulvophyceae. These groups are most described for the higher biomass in freshwater systems [56], exhibiting plankton biodiversity. Bacillariophyceae, the raphid diatoms, were found to be prominent in the gut profile of both river and weir chironomids. These diatoms have been reported to have conglomeration to live in the plankton community [57]. Olefeld et al. (2020) [58] stated that the diversity pattern in the freshwater system depends predominantly on diatoms (Bacillariophyceae), Ochrophyta (Chrysophyceae, Eustigmatophyceae), and fungi. Notably, we recorded diverse fungal groups in the diet profile (Table 3). Fungal content in the diet resembles the enzymatic synthesis linked to metabolizing activities in the gut [59,60] by converting detritus into more nutritious and readily digestible products [4,61].

### 4.3. Diet Contents Exhibiting the Habitat Condition

Abiotic factors are important drivers in the spatial distribution of the dominant communities [48,62]. The relationship between the environmental factors and species in an ecosystem is considered to be crucial. Environmental factors (conductivity, pH, water temperature, DO, turbidity, total nitrogen, chlorophyll-a) in the river and weir habitats showed a clustered pattern with phytoplankton and minor groups of fungi. Physiochemical factors in the aquatic ecosystem relate the geographical proximity to species occurrence and abundance [20]. Measurement of nutrients correlated with the occurrence and variability of fungal groups categorized (aerobic, anaerobic, facultative anaerobic). The enumerated dietary composition in the gut has been reported to have functional importance as planktonic/eukaryotic biodiversity in different ecosystems [18,19,21,63,64]. This was the preferential group in the chironomid diet, whereas the predominant algal populations could have a vast distribution from the bottom layers to the surface of the waters. An increased algal content in the water column leads to the preferred feeding by aquatic organisms among the other food sources [65]. Accordingly, the presence of diatoms matched the observations of Shirey et al. (2008) [66] that it is an indicative factor of eutrophic conditions, high biological oxygen demand, and low dissolved oxygen in the sediment layers. Considering Blazewicz-Paszkowycz and Ligowski, (2002) [67], the maximum microalgae-related species in the weir gut profile means there could be a possibility that chironomids feed on diatoms from the bottom zones in the water habitat. This positively induces the active role of chironomid feeding and food sources in the sediment layers of rivers and weirs. Likely, we identified diverse fungal groups as a plausible outcome. The fungal groups in the gut content typically determined the sediment conditions since the diverse fungal species in previous studies have been identified in the sediments of water habitats. This is a critical interpretation of the coexistence of the fungal population in planktons and their interaction distinctly in biogeochemical cycling in the water and sediment layers [61,68]. It was apparent that the presence of different planktonic fungal clusters denoted the involvement of fungi in the decomposition of organic matter and aggregation. A similar relation between fungi and sediments in the aquatic environment is inferred in other reports also [61,63,69,70]. The ratio of protozoa in this study symbolizes the feeding of chironomids at different layers in the habitats, even in the anoxic zones [53,71,72]. Gut profiling study implied a low distribution of zooplankton relating to biodiverse populations in the environment [27,73,74].

## 5. Conclusions

The multi-compositional diet profile of the *Polypedilum* larvae is an indication of the various food contents in the two contrasting habitats accomplished as the first attempt using eDNA metabarcoding of the V9 18S region. Altogether, the 29 different food content taxa in the *Polypedilum* larval gut were categorized into phytoplankton, fungi, protozoa, zooplankton, and others. Maximum coherence to food constituents corresponds to (i) phytoplankton as the most common food source (higher composition in IS and JS) with Chlorophyta as the abundant group, (ii) protozoa with more notable clusters in river samples than in weir, (iii) fungi—aerobic fungal groups were distributed significantly higher in both habitats with the maximum ratio in the diet of river chironomid (IS), and (iv) zooplankton—maximum frequency in the river chironomid, particularly high in SC.

The species richness with maximum OTU cluster occurrence and diversity was observed in the river chironomid (IS) as well as in GG and IP (weir chironomid). The functional OTUs certainly expressed the traits of the larval diet profile defined by dominant groups of Phytoplankton. The proportions of food items varied significantly between individual chironomids in both habitats. The larvae obtained from the river sites, especially from IS, revealed a generalized feeding reflected in the diet breadth values. Thus, demonstrating the heterogeneity in food content diversity with notable species spectra among river and weir habitats was made possible by eDNA metabarcoding.

## Figures and Tables

**Figure 1 ijerph-19-10945-f001:**
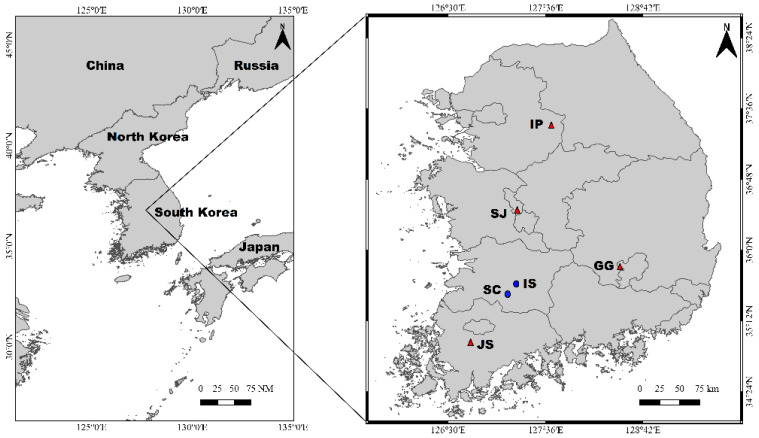
Sampling sites of the chironomid larvae from the rivers (SC—Sunchang and IS—Imsil, marked in blue dot) and weirs (IP—Ipo, SJ—Sejong, JS—Juksan, and GG—Gangjeong, marked in red triangle) in South Korea.

**Figure 2 ijerph-19-10945-f002:**
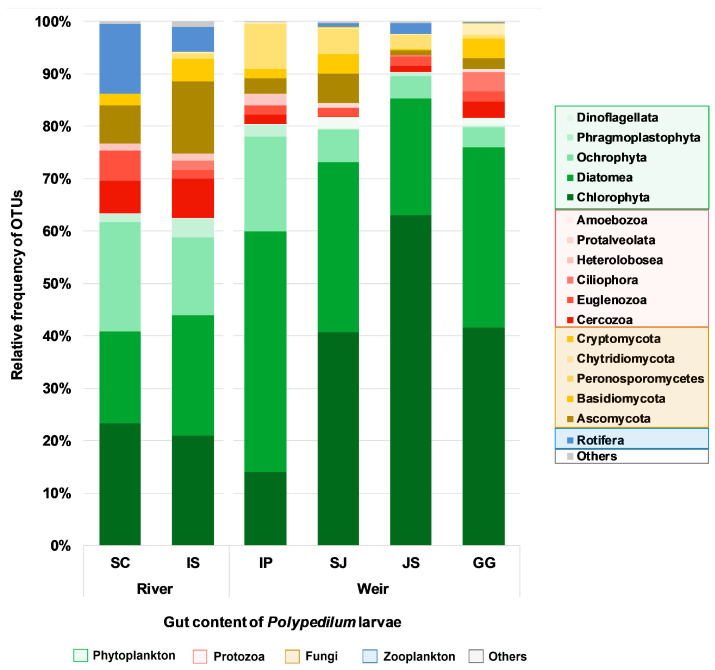
Relative OTU frequency of V9 18S rRNA sequence reads at phylum level showing major diet composition in the gut of chironomid. Unique/rare taxa comprising minor OTUs such as Annelida, Arthropoda, Cnidaria, Marine Stramenophiles, Mollusca, Nematozoa, Nucleariidae/Fonticula, Vertebrata, Neocallimastigomycota, Haptophyta (Prymnesiophyceae), Cryptophyta (Cryptophyceae) grouped as Others.

**Figure 3 ijerph-19-10945-f003:**
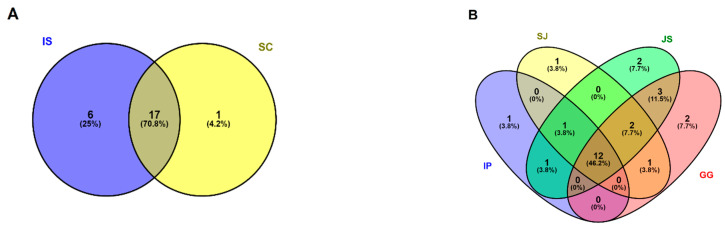
Venn diagram of shared and unique OTUs in the gut profile of river and weir midges. OTUs were clustered using a threshold distance of 0.03. (**A**). Further, 17 common taxa in IS and SC were Arthropoda, Ascomycota, Basidiomycota, Cercozoa, Chlorophyta, Ciliophora, Diatomea, Dinoflagellata, Euglenozoa, Heterolobosea, Nematozoa, Nucleariidae/Fonticula, Ochrophyta, Peronosporomycetes, Phragmoplastophyta, Rotifera and Vertebrata. 6 unique food sources of IS midges were Annelida, Chytridiomycota, Cryptophyta, Marine Stramenophiles, Mollusca, and Haptophyta and a rare group in SC is Cryptomycota. (**B**). 12 common food content in IP, SJ, JS, and GG were Ascomycota, Basidiomycota, Cercozoa, Chlorophyta, Chytridiomycota, Cryptomycota, Diatomea, Euglenozoa, Ochrophyta, Peronosporomycetes, Phragmoplastophyta, and Vertebrata. OTUs of Arthropoda were shared between IP, SJ, and JS. Rotifera was the shared food item in SJ, JS, and GG. Protalveolata was a common taxon in SJ and GG, Halobacterota in IP and JS, Ciliophora, Cnidaria, and Nucleariidae/Fonticula were the common diet components in JS and GG. Heterolobosea (IP), Annelida (SJ), Euryarchaeota, and Amoebozoa (JS) were the unique food items shown.

**Figure 4 ijerph-19-10945-f004:**
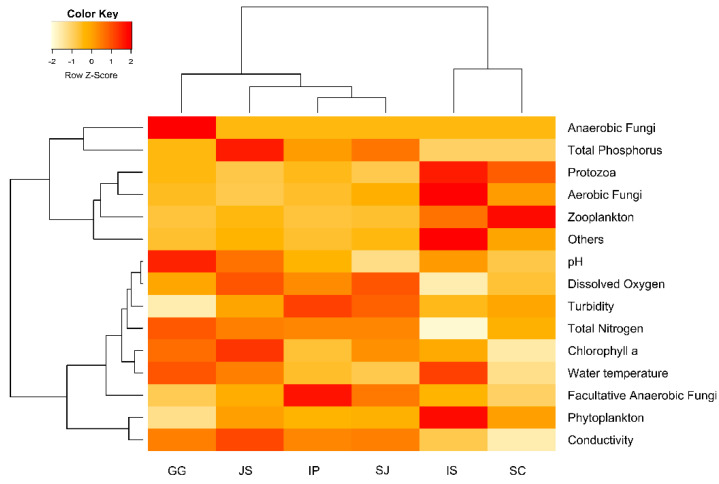
The two-way clustering analysis shows the relationship between environmental parameters with different food sources in the diet profile of river and weir chironomids. The vertical axis was the environmental conditions and major groups of food sources. The horizontal axis indicates the study habitats. Food sources—Phytoplankton was included with Chlorophyta, Diatomea, Dinoflagellata, Ochrophyta, Phragmoplastophyta, Cryptophyta, and Haptophyta; Aerobic Fungi (Ascomycota, Basidiomycota, Chytridiomycota, and Cryptomycota), Anaerobic Fungi (Neocallimastigomycota), Facultative Anaerobic Fungi (Peronosporomycetes); Amoebozoa, Cercozoa, Ciliophora, Euglenozoa, Heterolobosea, and Protalveolata were grouped to Protozoa, Annelida, Arthropoda, Cnidaria, Marine Stramenophiles, Mollusca, Nematozoa, Nucleariidae/Fonticula, and Vertebrata were grouped as others and Rotifera included as Zooplankton.

**Table 1 ijerph-19-10945-t001:** Oligonucleotide primer sequences with adapter regions (highlighted in bold) used to amplify the V9 18S rRNA gene of eukaryotic content from the *Polypedilum* larvae gut content.

Primer Name	Specificity	Primer Sequence (5′-3′)	Length (bp)	Reference
1380F	eukaryotic	GCCTCCCTCGCGCCATCAGXXXXXCCCTGCCHTTTGTACACAC	43	[17]
1510R	eukaryotic	GCCTTGCCAGCCCGCTCAGCCTTCYGCAGGTTCACCTAC	39	

**Table 2 ijerph-19-10945-t002:** Operational taxonomic units (OTUs), Good’s coverage, Chao1 (S), and niche breadth (B_A_) estimated for the diet composition of chironomid.

	Samples	OTUs	Good’s Coverage (%)	Chao Index (S)	Niche Breadth (B_A_)
River	SC	4066	100	6341.05	0.25
	IS	6361	100	8023.51	0.27
	IP	2004	100	2646.82	0.13
Weir	SJ	2031	100	2608.67	0.12
	JS	2014	100	2027.55	0.06
	GG	1372	100	3023.92	0.11

**Table 3 ijerph-19-10945-t003:** The fungal component of the gut contents of the *Polypedilum*.

Groups	Phylum	Class	River Samples		Weir Samples			
			SC	IS	IP	SJ	JS	GG
Aerobic	Ascomycota	Dothideomycetes	1.27	0.88	0.62	1	0.08	0.12
		Eurotiomycetes	0.84	5.76	0.01	0.09	0.07	0.07
		Leotiomycetes	0	0	0.01	0	0	0.08
		Saccharomycetes	0.01	0.44	0.04	0.09	0.04	0.03
		Sordariomycetes	0.12	0.41	-	-	-	0.01
		Pezizomycetes	-	-	-	0.01	-	-
		Unclassified Ascomycota	0.51	0.3	0.11	0.31	0.03	0.07
	Basidiomycota	Agaricomycetes	0.21	0.82	0.05	0.04	0.04	0.16
		Exobasidiomycetes	0.02	0.51	-	0.03	-	-
		Malasseziomycetes	0.29	0.48	0.27	0.32	-	0.3
		Microbotryomycetes	0.31	-	0.05	0.28	0.05	0.04
		Tremellomycetes	-	-	-	0.12	-	0.07
		Ustilaginomycetes	0.01	0.22	-	0.01	0.01	-
		Unclassified Basidiomycota	-	0.36	0.09	0.18	-	0.12
	Chytridiomycota	Chytridiomycetes	-	0.15	0.08	0.07	0.01	0.38
	Cryptomycota	Rozellidea	0.01	-	0.03	0.03	0.01	0.04
Subtotal			3.6	10.33	1.36	2.58	0.34	1.49
Facultative Anaerobic	Peronosporomycetes	Peronosporomycetes	-	0.58	2.28	1.31	0.7	0.13
Anaerobic	Neocallimastigomycota	Neocallimastigomycetes	-	-	-	-	-	0.03

The table indicates the fungal groups categorized as aerobic, facultative anaerobic, and anaerobic based on fungal taxon (Phylum: Class) and the relative frequency of operational taxonomic units (OTUs) at a distance of 0.03.

## Data Availability

Not applicable.

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
