# Peer review of "Profiling Analysis of Filter Feeder Polypedilum (Chironomidae) Gut Contents Using eDNA Metabarcoding Following Contrasting Habitat Types—Weir and Stream"

_ijerph, 2022, doi:10.3390/ijerph191710945_

Round 1

Reviewer 1 Report

The paper is of interest  considering the new approach to analyse diet composition of a Chironomid genus;

It is rich of information even if somewhat tedious, but surely rich of data.

Unfortunately there are no other contribution in literature to make comparisons,   but we hope that they will be available in the future.

My observations:

Pag 10 row 182-184 This sentence is not well formulated, it seems that chironomids belong to phytoplankton !

“River chironomid constituted 60% of phytoplankton representing 8 taxa in IS”

Figure 4

It seems that you carried out a cluster analysis with water bodies as objects (rows) and both environmental variables (temperature conductivity etc) and diet composition (phyto zoo-plankton etc) as attributes (columns); the results given in Fig. 4 are not quite easy to interpret;

in my opinion it should be of interest to carry out a constrained ordination between environmental variables and diet components,  with sites as objects

Pag 10 Table 4

The abundance of anaerobic fungi in GG can be related to oxygen condition of the site ?

It should be also of interest to know if only one species of Polypedilum is present and if other genera of Chironomids were present and why have you separated Polypedilum from other similar taxa genera

Reviewer 2 Report

The authors have contributed to science by investigating the gut contents Polypedilum (Chironomidae) from two different habitats (lentic and lotic) using eDNA metabarcoding a comparatively new technique. However, the manuscript isn’t publishable in its current form because the English requires further editing to improve clarity and grammatical correctness. I haven’t made specific comments past the abstract – included are examples of the changes required.

I question the value of some of the figures and tables – particularly those displaying the same/similar data in different ways e.g. Table 1 and Figures 2 and 3.

I’m not sure that the hierarchical clustering analysis clearly shows a relationship between the abiotic environmental factors and the chironomids’ diet. It would be worth exploring multivariate analysis (e.g. Non-Metric Multidimensional Scaling) and principal component analysis using a program such as PRIMER 7 & PERMANOVA+.

Review comments/suggestions

The inclusion of all the tables and figures generated needs to be re-evaluated.

I suggest minimizing the use of abbreviations (where possible) to improve readability of the text and interpretation of tables and figures.

Page 1, Line 11

……..larvae among collected from two contrasting….

Page 1, Line 12 Be consistent in the tense you use.

Our approach is was i) to apply eDNA-based……….

Page 1, Line 13

…….for revealing to identify the gut contents of the chironomid larvae,…

Page 1, Line 14. The sentence needs to be rewritten to clarify. I’m unsure of the appropriateness of the use of the term “ratiocination”. Do you mean the ratio of the different sediment types collected from the chironomid gut contents?

iii) ratiocination of habitats sediment condition with the food sources in the gut…..

Page 1, Line 15

The most consumed food most commonly found in the gut contents was Chlorophyta with…. Or

The most abundant food found in the gut contents was…….

Page 1, Lines 17 & 18. Are you still referring to the chironomid gut contents or the benthos?

The percentage of aerobic fungi in the river chironomids was 17 3.6 and 10.34 % (SC and IS), while it was in the range of 0.34 – 2.58 % in those collected from the weirs.

Page 1, Lines 24 & 25 Keywords

Chironomid gut contents,…….

River and weir habitats……..

Page 1, Lines 28 & 29 Introduction – needs to be rephrased

The first sentence needs to be rewritten:

Non-biting midges (Diptera: Chironomidae), are the most dominant groups in the aquatic environment and they are distributed abundantly in and around the aquatic habitats.

For example:

Non-biting midges (Diptera: Chironomidae), are one of the most abundant and widely distributed invertebrates in both lentic and lotic aquatic habitats.

Round 2

Reviewer 2 Report

Dear Editor/s

The manuscript has undergone a revision and the clarity has been improved. The authors have addressed all comments. They have made an important contribution to science by investigating the diet of chironomids collected from two different habitats using next generation sequencing.

Best wishes,

Anneke